A mathematical landmark-based method for measuring worn molars in hominoid systematics

Dykes Susan J. Sue.Dykes1@outlook.com 1 2
Pilbrow Varsha C. 3
1 Evolutionary Studies Institute, University of the Witwatersrand , Johannesburg , South Africa
2 School of Geosciences, University of the Witwatersrand , Johannesburg , South Africa
3 Department of Anatomy and Neuroscience, University of Melbourne , Melbourne , Australia
Hoover Kara
Electronic publication date: 2019 May 23
Publication date: 2019
Volume: 7
Electronic Location ID: e6990
Received 2019 Feb 15; Accepted 2019 Apr 20
Copyright: ©2019 Dykes and Pilbrow
Copyright year: 2019
Copyright holder: Dykes and Pilbrow
License: This is an open access article distributed under the terms of the Creative Commons Attribution License, which permits unrestricted use, distribution, reproduction and adaptation in any medium and for any purpose provided that it is properly attributed. For attribution, the original author(s), title, publication source (PeerJ) and either DOI or URL of the article must be cited.
License URL: https://creativecommons.org/licenses/by/4.0/

Keywords: Dental morphometrics, Worn teeth, Taxonomy, Systematics, Landmarks, Geometric morphometrics

Funding: National Research Foundation of South Africa 102169 DST-NRF Centre of Excellence in Palaeosciences D2015/01SD Palaeontological Scientific Trust (PAST) LSB Leakey Foundation, National Science Foundation SBR-9815546 Wenner-Gren Foundation Susan J. Dykes received financial support from the National Research Foundation of South Africa (grant number 102169), the DST-NRF Centre of Excellence in Palaeosciences (D2015/01SD) and the Palaeontological Scientific Trust (PAST). Varsha C. Pilbrow received grants from the LSB Leakey Foundation, National Science Foundation (SBR-9815546), and the Wenner-Gren Foundation. The funders had no role in study design, data collection and analysis, decision to publish, or preparation of the manuscript.

==============================
Worn teeth pose a major limitation to researchers in the fields of extinct and extant hominoid systematics because they lack clearly identifiable anatomical landmarks needed to take measurements on the crown enamel surface and are typically discarded from a study. This is particularly detrimental when sample sizes for some groups are already characteristically low, if there is an imbalance between samples representing populations, sexes or dietary strategies, or if the worn teeth in question are type specimens of fossil species or other key specimens. This study proposes a methodology based predominantly on mathematically-derived landmarks for measuring size and shape features of molars, irrespective of wear. With 110 specimens of lower second molars from five species of extant hominoids (Pan troglodytes, P. paniscus, Gorilla gorilla, G. beringei, Homo sapiens), n ≥ 20 per species, n ≥ 10 per subspecies, good species separation in morphospace is achieved in a principal components analysis. Classification accuracy in a discriminant function analysis is 96.4% at the species level and 88.2% at the subspecies level (92.7% and 79.1%, respectively, on cross-validation). The classification accuracy compares favorably to that achieved by anatomically-derived measurements based on published research (94% and 84% at the species and subspecies level respectively; 91% and 76% on cross-validation). The mathematical landmarking methodology is rapid and uncomplicated. The results support the use of mathematical landmarks to enable the inclusion of worn molar teeth in dental studies so as to maximize sample sizes and restore balance between populations and/or sexes in hominoid systematic studies.

Introduction

Studies into the population systematics of extant hominoids are of great significance to primatologists, anthropologists and paleoanthropologists alike. Primatologists utilize such studies of diversity to obtain a better understanding of the natural history of the hominoids and promote conservation of endangered groups (for example, Wolfheim, 1983; Oates, 1996; Butynski, 2003; Kalpers et al., 2003; Kormos et al., 2003; Taylor & Goldsmith, 2003; Bergl, 2006; Oates et al., 2007; Williamson & Fawcett, 2008; Plumptre et al., 2010; Junker et al., 2012; Nater et al., 2017). Anthropologists and paleoanthropologists study species-subspecies diversity to provide analogues upon which to base conclusions regarding alpha taxonomy and the naming of new species in the fossil hominin context (Vitzthum, 1984; Ferguson, 1989; Wood, Li & Willoughby, 1991; Uchida, 1992; Uchida, 1996; Albrecht & Miller, 1993; Shea, Leigh & Groves, 1993; Richmond & Jungers, 1995; Braga, 1995; Plavcan & Cope, 2001; Albrecht, Gelvin & Miller, 2003; Pilbrow, 2003; Braun, Thackeray & Loots, 2004; Mitteroecker et al., 2004; Scott & Lockwood, 2004; Lee, 2005; Baab, 2008; Lordkipanidze et al., 2013). Extant hominoid species that are closely related to extinct hominin species are considered to be valuable proxies or analogues of morphological variability in fossil hominin species (Kimbel & Martin, 1993; Ackermann, 2003).

An understanding of variability within and between species and subspecies in extant hominoid groups is therefore key to predicting how much variability to expect within and between fossil hominin species. As sample sizes for fossil hominin skeletal elements are limited, many studies focus on teeth, which are the most abundantly-represented skeletal element in the fossil record due to the excellent preservation qualities of the outer enamel surface (OES). Apart from the fact that teeth are likened to preformed “fossils” (Boyde, 1997, p. 29), in that, once erupted, they do not change size or shape during the individual’s lifetime, making them a useful resource for morphometric analyses, their value is enhanced by the fact that large samples of teeth with good provenance and sex data are available in museums.

Unworn or minimally worn teeth lend themselves to many different types of three-dimensional (3D) and two-dimensional (2D) analyses. Micro-computerized tomography scans of teeth allow for 3D studies to be carried out into the enamel-dentine junction (EDJ), which preserves in pristine form the underlying morphology of the dentine. The EDJ provides sufficient information with which to differentiate between species (Skinner et al., 2008; Skinner et al., 2009; Braga et al., 2010; Crevecoeur et al., 2014). The sharpness of the ridges running between the dentine horns provides secure and repeatable landmarking sites (Skinner et al., 2009). Nevertheless, the EDJ proves difficult to landmark once wear has progressed through the enamel, obliterating the dentine horns. In such situations the dentine peaks either need to be inferred or such specimens are typically not selected for CT scanning. Some further challenges for 3D studies of teeth include the high cost of scanning, difficulties in segmenting enamel from dentine in highly fossilized teeth, and the time involved in segmenting dental tissues from image stacks.

The use of 2D imagery still remains attractive, as geographically-comprehensive collections of images may be obtained relatively swiftly and cost-effectively from museums around the world and amalgamated into manageable databases. Shape and size analyses of the morphology of the OES, as studied from occlusal-view photographs of molars, have been successful in discriminating between extant great ape species, subspecies and even populations (Pilbrow, 2003; Pilbrow, 2006; Pilbrow, 2010; Uchida, 1998a; Uchida, 1998b; Uchida, 2004; Singleton et al., 2011). This type of image has proven equally useful for fossil hominin taxonomic studies (e.g., Wood & Abbott, 1983; Wood, Abbott & Graham, 1983; Suwa, Wood & White, 1994; Suwa, White & Clark Howell, 1996; Suwa, 1996; Bailey, 2004; Martinón-Torres et al., 2006; Gómez-Robles et al., 2007; Gómez-Robles et al., 2008; Gómez-Robles et al., 2012; Gómez-Robles et al., 2015). The landmarking process tends to be relatively quick, accurate and cost effective. However, once past the first stages of wear, anatomical landmarks such as cusp tips, crests, and foveae are either difficult to identify or are obliterated from view completely. The researcher usually discards such specimens, resulting in reductions in sample sizes (see also Stojanowski & Johnson, 2015). This is problematic in cases where the specimens are holotypes or paratypes of fossil species and should warrant inclusion, or where fossil sample sizes are generally low, or there is an imbalance between specimens representing males and females, geographical regions or dietary strategies.

The purpose of this paper is to propose a method for including worn molar crowns in taxonomic studies by capitalizing on the few anatomical landmarks that remain discernible even after considerable wear. These anatomical landmarks, which are classified as Type I (Bookstein, 1991), are sited at the grooves separating the main cusps at the perimeter of the molar crown. The grooves are usually visible despite high levels of wear on the enamel surface. We start with these Type I landmarks and thereafter use mathematically-derived Type III landmarks (Bookstein, 1991) at strategic points on and around the crown. Thereafter, landmarks are used to calculate linear and angular measurements, which provide detailed information on the shape of the tooth and serve as the raw data for further analyses. We test the hypothesis that measurements obtained from mathematically-derived landmarks provide at least as good discrimination between chimpanzee, gorilla and human molars as obtained from anatomically-derived landmarks in 2D and 3D studies. We use lower second molars in this study, but the methodology is applicable to other molar types.

Materials and Methods

Materials

We selected 110 occlusal-view 2D images of lower second molars (LM2) to represent five species (eight subspecies) of extant hominoids: Gorilla beringei beringei (n = 10), G. b. graueri (n = 10), G. gorilla gorilla (n = 20), Pan troglodytes verus (n = 10), P. t. troglodytes (n = 10), P. t. schweinfurthii (n = 10), P. paniscus (n = 20) and modern Homo sapiens (n = 20, of which 8 were selected to represent hunter-gatherer groups and 12 to represent groups with other subsistence strategies). The photographs relating to Pan and Gorilla were randomly selected from the images used by Pilbrow (2003), Pilbrow (2006), Pilbrow (2007) and Pilbrow (2010), and those relating to Homo sapiens were randomly selected from the images used by Dykes (2018). The selected samples were equally balanced between males and females to ensure that variation from sexual dimorphism was represented and chosen from geographically-diverse populations to represent inter-population variation. All teeth included in this study come from previous studies and were at minimal stages of wear, with either no dentine exposed, dentine exposed as points on cusp tips, or dentine exposed as small pits on cusp tips (Pilbrow, 2003). The reason for utilizing relatively unworn teeth in this study is to allow for the methodology to be compared for accuracy with existing methodologies, which are reliant on identifying anatomical landmarks. Nevertheless, it is important to reiterate that the landmarking method presented here remains valid for both worn and unworn teeth, because the landmarks atop the surface of the enamel are calculated identically, whether or not occlusal surface features still remain visible. This holds true of most stages of advanced tooth wear, provided that the perimeter wall (the occlusal outline in each image) is still intact with cusp intersections along the perimeter edge visible. A list of images used is summarized in Table 1.

Table 1 Summary of images of 110 LM2s used in the study.

Species/subspecies	Population name/geographical areab	Number/Sex	Collections useda	
G. b. beringei (Eastern mountain gorillas)	Virunga; Kayonza	F = 5; M = 5	RG; USNM; BMNH	
G. b. graueri (Eastern lowland gorillas)	Utu; Mweng-Fizi; Tshiaberimu	F = 5; M = 5	RG	
G. g. gorilla (Western lowland gorillas)	Coastal Cameroon; Coastal Gabon; Southern Gabon, Congo; Sangha River; Upper reaches of Sangha and Sanaga Rivers; Inland Cameroon	F = 10; M = 10	ZMB; BMNH; Z; PCM; USNM; RG; MCZ	
P. t. verus (Western chimpanzees)	Between Gambia and Cavally; between Cavally and Volta	F = 5; M = 5	RG; USNM; PM; AMNH	
P. t. troglodytes (Central chimpanzees)	South Sanaga River; Sanaga River, inland of coast; Southern Gabon	F = 5; M = 5	Z; PCM; MCZ; RG; ZMB	
P. t. schweinfurthii (Eastern chimpanzees)	Between Ubangi and Congo-Lisala; Uele River; Kisangani district; Lake Albert to north of Lake Tanganyika; Lake Kivu and Lake Tanganyika	F = 5; M = 5	RG; MBNH; ZMB;	
P. paniscus (Bonobos)	Between Congo and Lukenie; between Lomami and Congo; between Lukeni and Kasai	F = 10; M = 10	RG; MCZ	
H. sapiens (Recent modern humans)	Southern African KhoeSan (hunter-gatherer); Kenya - Babinga (hunter-gatherer); Kenya - Teita (subsistence farming); Australian Aboriginal (hunter-gatherer); Melanesia (horticulturalists); South Asia (predominantly agriculturalists); Balkan region (predominantly agriculturalists); Near East (predominantly agriculturalists); Western Europe (predominantly agriculturalists)	F = 10; M = 10	Iziko; Dartc; MNHN; CAM-DL	
Notes.

a Museum/collections are as follows: AMNH, American Museum of Natural History, New York; BMNH, British Museum of Natural History, London; CAM-DL, Duckworth Collection, Cambridge University; Dart, Raymond Dart Collection, Anatomical Sciences, University of the Witwatersrand, Johannesburg; Iziko, Iziko Museum, Cape Town; MNHN, Muséum National d’Histoire Naturelle, Paris (Musée de l’Homme) ; PCM, Powell-Cotton Museum, Kent ; RG, Musée Royal de L’Afrique Centrale, Tervuren, Belgium; USNM, United States National Museum, Washington, D.C.; Z, Anthropologisches Institüt und Museum der Universität Zürich-Irchel, Zürich; ZMB, Zoologisches Museum, Berlin.

b Pilbrow (2006); Pilbrow (2010)

c University of the Witwatersrand, Johannesburg, Blanket Ethics Waiver Number W-CJ-14064-1.

Image processing

All photographs were taken by the authors, using identical methodology, as described in Bailey, Pilbrow & Wood (2004), Pilbrow (2006), Pilbrow (2010), and Dykes (2014) and Dykes (2018). Images were then processed using GIMP® (the freeware equivalent of Adobe Photoshop®) as left-side teeth (right-side teeth were mirrored if necessary, in keeping with other concurrent research projects) with the mesial side of the tooth to the left of the image, the distal side to the right, the lingual side to the top of the image and the buccal side to the bottom. The mesial edge and the mesiodistal groove of the tooth in normal rotation (Goose, 1963) provides a guideline for the longitudinal axis of the tooth (Wood, 1991; Benazzi et al., 2012), and this is oriented horizontally on screen. Corrections of any interstitial wear or slight damage to the perimeter outline of the tooth are carried out digitally in Adobe Illustrator® following the methodology of Wood & Abbott (1983). Thereafter, the rectangle tool was used to superimpose a bounding box around the perimeter of the molar, to stand proxy for the corrected mesiodistal (MD) and maximum buccolingual diameter (BL) measurements, the latter being at right angles to the MD diameter (Wood & Abbott, 1983). The bounding box can be seen in Fig. 1.

Figure 1 Landmarks sited on a Pan troglodytes lower second molar (RG, Tervuren, #29075).

Table 2 Description of landmark sites.

#	Type	Description	
1	III	Mathematical center of bounding box	
2	III	Mesial-most extent of the molar, placed midway down the mesial side of the bounding box	
3	III	Lingual-most extent of the molar, placed midway across the lingual side of the bounding box	
4	III	Distal-most extent of the molar, placed midway down the distal side of the bounding box	
5	III	Buccal-most extent of the molar, placed midway across the buccal side of the bounding box	
6	I	Anatomical landmark at the groove between the metaconid and protoconid on the perimeter of the crown (corrected for interstitial wear).	
7	I	Anatomical landmark at the groove between the metaconid and entoconid on the perimeter of the crown.	
8	I	Anatomical landmark at the groove between the protoconid and hypoconulid on the perimeter of the crown.	
9	I	Anatomical landmark at the groove between the hypoconulid and hypoconid on the perimeter of the crown.	
10	I	Anatomical landmark at the groove between the hypoconid and protoconid on the perimeter of the crown.	
11	III	Midpoint of the line between landmarks 6 and 7.	
12	III	Midpoint of the line between landmarks 7 and 8.	
13	III	Midpoint of the line between landmarks 8 and 9.	
14	III	Midpoint of the line between landmarks 9 and 10.	
15	III	Midpoint of the line between landmarks 10 and 6.	
16	III	Midpoint of the arc/curve at the perimeter of the metaconid created by extending a straight line from landmark 1 through landmark 11 to the perimeter.	
17	III	Midpoint of the arc at the perimeter of the entoconid created by extending a straight line from landmark 1 through landmark 12 to the perimeter.	
18	III	Midpoint of the arc at the perimeter of the hypoconulid created by extending a straight line from landmark 1 through landmark 13 to the perimeter.	
19	III	Midpoint of the arc at the perimeter of the hypoconid created by extending a straight line from landmark 1 through landmark 14 to the perimeter.	
20	III	Midpoint of the arc at the perimeter of the protoconid created by extending a straight line from landmark 1 through landmark 15 to the perimeter.	
21	III	Mathematically-derived proxy for the center point of the metaconid, placed at the midpoint of the line from landmark 1 to landmark 16	
22	III	Mathematically-derived proxy for the center point of the entoconid, placed at the midpoint of the line from landmark 1 to landmark 17	
23	III	Mathematically-derived proxy for the center point of the hypoconulid, placed at the midpoint of the line from landmark 1 to landmark 18	
24	III	Mathematically-derived proxy for the center point of the hypoconid, placed at the midpoint of the line from landmark 1 to landmark 19	
25	III	Mathematically-derived proxy for the center point of the protoconid, placed at the midpoint of the line from landmark 1 to landmark 20	
26	III	Point on the lingual edge of metaconid placed by extending a line from the center point of metaconid to the lingual edge of the tooth.	
27	III	Point on the lingual edge of entoconid placed by extending a straight line from the center point of entoconid to the lingual edge of the tooth.	
28	III	Point on the buccal edge of hypoconid placed by extending a straight line from the center point of hypoconid to the buccal edge of the tooth.	
29	III	Point on the buccal edge of protoconid placed by extending a straight line from the center point of protoconid to the buccal edge of the crown.	

Landmarking

In total, 29 landmarks were chosen to represent the general dimensions, key points around the occlusal perimeter and the cusp arrangements of the tooth. The landmarks are depicted in Fig. 1 and described in Table 2. These landmarks also allowed for easily-interpretable wireframes to be produced for the analysis of relative warps in a Principal Components Analysis (PCA), and to characterize consensus molar shapes of the groups. Measurements taken between landmarks were also adaptable for further discriminant function and other statistical analyses. The first landmark (1) was placed at the geometric center of the tooth as calculated from the bounding box. Four further landmarks (2, 3, 4, 5) were placed around the perimeter of the box to mark the corrected mesiodistal and the maximum buccolingual diameter of the tooth. The next five landmarks (6, 7, 8, 9, 10) are Type I anatomical landmarks positioned at the points where the grooves between the cusps intersect with the perimeter of the tooth. These are the only Type I landmarks used in this study. Five additional landmarks (11, 12, 13, 14, 15) were placed at the midpoints of the lines connecting the anatomical landmarks, 6 –10. These helped to provide a general orientation of each cusp. The next five landmarks (16, 17, 18, 19, 20) were placed at the edge of the crown to mark the center point of each cusp arc around the perimeter. These were identified by drawing straight lines from the bounding box center (from landmark 1) to the edge of the crown while bisecting the lines connecting the Type I landmarks. Thus, a line from the center passed through landmark points 11–15 to reach the perimeter of the crown and provide a landmark point. The midpoints of the lines from landmark 1 to the peripheral landmarks 16 –20 themselves formed an additional five landmarks (21–25), which were used to stand proxy as the mathematical center of each cusp. Finally, the mathematically-derived centers of the mesial and distal cusps, respectively, metaconid and protoconid, and entoconid and hypoconid, were used in extending lines towards the lingual and buccal edge of the crown and provide four additional peripheral landmarks (26–29). These helped to provide an orientation of the mesial and distal cusps relative to the longitudinal axis of the crown. In all, 14 landmarks were used to provide a wireframe outline shape of the tooth: five pinpointing the cusp intersections, five marking the mathematically-derived centers of each cusp arc and four locating the orientations of the mesial and distal cusps relative to the longitudinal orientation of the crown. Features on the surface of the crown were captured in wireframes by two polygons: an outer polygon joining the five cusp intersections at the periphery, and an inner polygon formed by the five mathematical midpoints of each cusp. Landmark 1 is identified in the wireframe via the inclusion of the MD and BL diameters, which intersect at the center of the tooth in the occlusal basin of the tooth.

Landmarking was carried out using ImageJ® freeware, which has the capacity to scale images and which has a line segment tool that shows the midpoint of lines traced onto the image and a “blob” tool to mark these midpoints with a colored dot. The landmark placement tool outputs the x and y coordinates of landmarks after they have been sited, ready for export to any spreadsheet software such as Microsoft Excel®, which may be prepopulated with formulae to calculate distances and angles between landmarks. The whole process of scaling, marking midpoints and landmarking each tooth takes on average three to four minutes to complete.

Special landmarking cases—molar crowns with four or six cusps

All molars in a landmark-based analysis require the same number of landmarks per specimen. For teeth with four cusps, for instance in certain individuals of modern H. sapiens, a modification is made to the landmarks pertaining to the absent hypoconulid (landmarks 13, 18 and 23). In this situation a hypoconulid is inferred from the small groove separating the entoconid and hypoconid (Fig. 2). Landmarks are sited on this small inter-cusp area as though it were a normal hypoconulid.

Figure 2 Landmarks on H. sapiens LM2 with four cusps.

In the case of a lower molar with six cusps (C6 or tuberculum sextum), the C6 is bisected between the entoconid and hypoconulid for purposes of marking the intersection between these cusps (Wood, 1991, p. 306, Fig 8.13[j]). This allows the same number of landmarks to be maintained across specimens (Fig. 3).

Figure 3 Homo sapiens LM2 with six cusps.

Intra-observer and inter-observer errors

Intra- and inter-observer errors were measured for tilt of molars at the image-capturing stage and orientation of the on-screen image at the image-processing stage. To quantify intra-observer errors of tilt, Amira® software was used to analyze differences in tilt angles of the occlusal surface in the x, y and z planes of three different images of the same tooth, all taken on separate occasions. The maximum difference in tilt between these images was 0.014 degrees along the x plane, 0.107 degrees along the y plane and 0.098 degrees along the z plane. To calculate the effect of such errors of tilt, landmarks were placed on the surface of a 3D image of a pristine tooth and Amira® software was used to measure landmark coordinate changes at various degrees of tilt across the buccolingual axis (the y plane, where tilt is most likely to occur during the photographic process). It was found that an error of tilt at 2 degrees would affect the landmark coordinate placements by 1% over the length of the buccolingual axis. Inter-observer error in the longitudinal rotation of molars during the image-processing stage was evaluated by two observers using five teeth randomly selected from five different species over a period of approximately six months. Landmarks were placed on each image and the mean deviations between these coordinates were calculated. The average deviation measured against the length of the mesiodistal axis was 0.295%, and the average deviation measured against length of the buccolingual axis was 0.316%.

Principal Component Analysis (PCA)

After translation, rotation and scaling of the images via a Generalized Procrustes Analysis (GPA), two types of principal component analyses were conducted in Morphologika®: traditional shape-only or shapespace PCA and size-versus-shape or formspace PCA (Mitteroecker et al., 2004; Mitteroecker et al., 2013). This latter PCA adds the natural logarithm of the centroid size for each specimen as a variable in the analysis and the resultant plot shows predominantly size variation along the first principal component, with the smallest specimens grouping at the negative end of the axis and the largest specimens at the positive end. A formspace PCA is particularly useful in taxonomic analyses with molars, because tooth size, which remains unchanged after the tooth has erupted, can be an important diagnostic feature in interspecific and intraspecific analyses. Morphologika® also calculates wireframes (or point clouds) and a slider bar allows these wireframes to be shown as relative warps along the x and y axes, for immediate visualization of shape changes along each principal component axis. These relative warps wireframes provide useful interpretation of PCA plots.

Discriminant function analyses (DFA)

The same sample of 110 hominoid lower second molars was analyzed by means of stepwise discriminant function analyses (DFA) in SPSS®, with leave-one-out cross-validation. A DFA minimizes within-group variation and maximizes between-group variation, providing good understanding of relative separation among the groups being analyzed. In a stepwise DFA, variables are included in the analysis until they no longer provide any further significant discrimination between groups, at which point redundant variables are removed from the analysis (Manly, 2005). Euclidean coordinates of landmarks are used to derive measurements for the DFA. Depending on the analysis and the sample sizes, these can include the natural log of centroid size as a proxy for overall tooth size, linear dimensions of the tooth crown, orientation of occlusal features measured in radians of angles, and shape features measured as ratios between landmarks, as shown in Fig. 4. For the present analysis, as the minimum sample size per group was 10 individuals, nine measurements that provided the highest canonical loadings were selected for the DFA. These are described in Table 3.

Figure 4 Raw distances (A) and angles (B) for use in DFA analyses.

Table 3 List of mathematically-derived measurements used for input into discriminant function analyses in this study.

#	Measurement	Description	
1	Mesiodistal (MD) diameter	Between landmarks 2 and 4	
2	Buccolingual breadth across mesial cusps	Between landmarks 26 and 29	
3	Buccolingual breadth across distal cusps	Between landmarks 27 and 28	
4	Breadth across buccolingual groove	Between landmarks 7 and 10	
5	Length of mesial edge of buccal development groove	Between landmarks 29 and 10	
6	Length of distal edge of buccal development groove	Between landmarks 10 and 28	
7	Angle of mesial cusps	Angle between line connecting centers of mesial cusps (landmarks 21 and 25) and the MD diameter (line between landmarks 2 and 4)	
8	Angle of distal cusps	Angle between line connecting centers of distal cusps (landmarks 22 and 24) and the MD diameter (line between landmarks 2 and 4)	
9	Hypoconulid curvature ratio	The extent of the outward projection of the of the arc of the hypoconulid at the perimeter, in relation to the total length between the tooth center and the midpoint of the hypoconulid at the perimeter, to measure flatness or curvature of the hypoconulid (landmarks 13 –18, divided by landmarks 1–18)	

Testing the accuracy of the methodology against traditional (anatomically-based) methodologies

Stepwise DFA with leave-one-out cross-validation was also used to test whether mathematically-derived measurements would produce the same level of classification accuracy as anatomically-derived measurements. To do this, we used the same 110 specimens chosen for the other analyses, to provide identically-matched samples. Anatomically-derived measurements from previous publications (Pilbrow, 2006; Pilbrow, 2010) were selected for the 90 chimpanzee and gorilla LM2s used in this study. The 20 human LM2 anatomical measurements were taken anew using the same molar images as this study. As the sample sizes per group in this study are smaller than in the previously published studies, a smaller set of independent variables were selected for the stepwise DFA to ensure that we met the assumptions of a robust DFA. The following nine variables that provided the highest canonical loadings were selected: length of crown, breadth of crown measured at mesial and distal cusps, distance between mesial and distal cusps, orientation of buccal and lingual cusps, and orientation of hypoconulid and cristid obliqua (Pilbrow, 2006).

In a further comparison of the classification accuracy of mathematical landmarks with anatomical landmarks, a random sample of 25 specimens was chosen, simulating the species-subspecies groups used in a study by Skinner et al. (2009) on discriminating species and subspecies of Pan using EDJ morphology. Thus, specimens of P. t. troglodytes, P. t. verus and P. paniscus were selected. Classification accuracy was computed using the same nine mathematically-derived variables as described above and compared with the results from Skinner et al. (2009). Bivariate plots along the first two discriminant functions were also compared for grouping patterns.

Results

Principal components analyses

The first two principal components in the shapespace (shape only) analysis are shown in Fig. 5. Relative warps wireframes at the ends of each axis show the average molar shape change along that axis.

Figure 5 Principal Components analysis in shapespace (shape only) of 5 extant hominoid species.

Legend: open circles, Gorilla gorilla; closed diamonds, G. b. beringei; X-crosses, G. b. graueri; T, Pan troglodytes troglodytes; S, P. t. schweinfurthii; V, P. t. verus; Targets, P. paniscus; Stars, H. sapiens. Red symbols denote females, blue symbols denote males. All wireframes depict molars with the mesial edge to the left, the distal edge to the right, the lingual edge to the top and the buccal edge to the bottom.

Pan, H. sapiens and Gorilla separate well in morphospace, but species and subspecies of Pan overlap with each other, as do species and subspecies of Gorilla. PC1 accounts for 66.1% of variance and relative warps wireframes indicate that broad teeth with distally-oriented hypoconulids and buccodistally-oriented hypoconids group towards the negative end of the x-axis (Pan and H. sapiens), while relatively narrow teeth with buccodistally-oriented hypoconulids and buccally-oriented hypoconids (Gorilla) group towards the positive end. PC2 (y-axis) accounts for 11.6% of variance, with broad molars, having reduced or absent hypoconulids, grouping towards the negative end of the axis (certain H. sapiens molars), and narrow molars with larger, well-defined hypoconulids grouping towards the positive end of the axis. Most Pan molars plot above the x-axis, with the exception of a few P. t. verus molars that are slightly broader across the crown and plot below the x-axis. Separation in morphospace is therefore good at the level of genus, but lacking at the species and subspecies level. There is also no separation in shapespace between molars belonging to male and female gorillas, indicating that in the absence of a size component, there is little to determine sexual dimorphism in shape alone.

In the size-versus-shape (formspace) analysis (Fig. 6), with size added back as a variable, small molars group towards the negative end of the x-axis and large molars group towards the positive end (PC1). This first component accounts for 93.1% of variance and PC2 (y-axis) accounts for 2.3% of variance. In formspace, P. troglodytes, P. paniscus and H. sapiens now separate well and can be differentiated spatially. Species and subspecies of Gorilla still overlap with each other, but not as much as in the shape-only PCA. The molars of G. b. graueri group as generally larger than molars of G. b. beringei, which in turn group as generally larger than those of G. gorilla. Males with larger molars tend to group at more positive values along the axis, although the separation between males and females is most discernable in the sexually dimorphic gorillas. Along PC2 (the y-axis), Gorilla species mostly plot above the axis (narrower teeth) with a few molars grouping just below the x-axis (slightly broader across the crown). Examining the plot and the wireframes, small, relatively narrow molars with pronounced hypoconulids group in the top left quadrant of the graph, with Pan paniscus well separated to the top left-hand side of the plot, being the smallest molars in the sample. Pan troglodytes molars generally group in this quadrant as well, particularly those belonging to P. t. troglodytes and P. t. schweinfurthii, which show considerable overlap. In the bottom left quadrant, relating to molars that are progressively broader across the crown, H. sapiens is generally separated from the few P. troglodytes molars notably those belonging to P. t. verus, which are generally relatively broader across the crown than are specimens from the other two Pan subspecies. In addition to the relative broadness of the crowns, the hypoconulid becomes less pronounced in the individuals grouping towards negative values of PC2.

Figure 6 Principal Components analysis in formspace (shape-versus-size) of 5 extant hominoid species.

Legend: open circles, Gorilla gorilla; closed diamonds, G. b. beringei; X-crosses, G. b. graueri; T, Pan troglodytes troglodytes; S, P. t. schweinfurthii; V, P. t. verus; Targets, P. paniscus; Stars, H. sapiens. Red symbols denote females, blue symbols denote males. All wireframes depict molars with the mesial edge to the left, the distal edge to the right, the lingual edge to the top and the buccal edge to the bottom.

DFA Classification accuracy, 110 LM2s from five species/eight subspecies

Stepwise discriminant function analyses showed that classification accuracy at the species level was 96.4% (Table 4). Classification accuracy for individual groups were as follows: G. beringei (90%); G. gorilla (95%); P. troglodytes (96.7%), P. paniscus (100%) and H. sapiens (100%). Two of the twenty G. beringei specimens classified with G. gorilla, with one G. gorilla reciprocally classifying with G. beringei. Within Pan, one P. troglodytes (n = 30) was grouped with P. paniscus. On cross validation, one further G. gorilla and G. beringei molar each classified reciprocally, one P. paniscus specimen grouped with P. troglodytes and one H. sapiens molar classified with P. paniscus, bringing the cross-validated classification accuracy to 92.7%. The coefficients most influencing the analysis along Function 1, which accounted for 91.9% of variance, were the buccolingual groove measurement (negatively loaded) and the distal cusp measurement (positively loaded). Other loadings contributing to discrimination between groups along Function 1 were the breadth measurement across the mesial cusps (positively loaded), the distal edge of the buccal development groove (negatively loaded), the mesiodistal measurement, and the angle of the mesial cusps (negatively loaded): this angle is juxtaposed against the angle of the distal cusps, which is positively loaded. Size, particularly breadth across the tooth, is therefore the main discriminating feature at the species level, but it is noted that the relationship between the mesial cusps and the distal cusps plays a major role (linear measurements and angles between the two sides, and the distance of the distal cusps from the buccal side of the buccolingual groove). Along Function 2, which accounted for a further 6.6% of variance, discriminating factors include the breadth measurement along the buccolingual groove, the ratio of the curvature of the hypoconulid in relation to its length from the tooth center, followed by the breadth of the mesial cusps.

Table 4 Classification accuracy of 110 LM2s at species level using mathematically-derived measurements.

	Species	G. beringei	G. gorilla	P. troglodytes	P. paniscus	H. sapiens	N	
Original	G. beringei	18	2	0	0	0	20	
G. gorilla	1	19	0	0	0	20	
P. troglodytes	0	0	29	1	0	30	
P. paniscus	0	0	0	20	0	20	
H. sapiens	0	0	0	0	20	20	
Cross-validated	G. beringei	17	3	0	0	0	20	
G. gorilla	2	18	0	0	0	20	
P. troglodytes	0	0	29	1	0	30	
P. paniscus	0	0	1	19	0	20	
H. sapiens	0	0	0	1	19	20	
96.4% of original grouped cases correctly classified.	
92.7% of cross-validated grouped cases correctly classified.	

Table 5 Classification accuracy of 110 LM2s at subspecies level using mathematically-derived measurements.

	Subspecies	G. b. beringei	G. b. graueri	G. g. gorilla	P. t. verus	P. t. troglodytes	P. t. schweinfurthii	P. paniscus	H. sapiens	N	
Original	G. b. beringei	8	1	1	0	0	0	0	0	10	
G. b. graueri	2	7	1	0	0	0	0	0	10	
G. g. gorilla	1	1	18	0	0	0	0	0	20	
P. t. verus	0	0	0	9	1	0	0	0	10	
P. t. troglodytes	0	0	0	0	8	1	1	0	10	
P. t. schweinfurthii	0	0	0	0	2	8	0	0	10	
P. paniscus	0	0	0	0	1	0	19	0	20	
H. sapiens	0	0	0	0	0	0	0	20	20	
Cross-validated	G. b. beringei	6	3	1	0	0	0	0	0	10	
G. b. graueri	3	6	1	0	0	0	0	0	10	
G. g. gorilla	1	1	18	0	0	0	0	0	20	
P. t. verus	0	0	0	9	1	0	0	0	10	
P. t. troglodytes	0	0	0	0	7	2	1	0	10	
P. t. schweinfurthii	0	0	0	1	4	5	0	0	10	
P. paniscus	0	0	0	0	1	0	19	0	20	
H. sapiens	0	0	0	2	1	0	0	17	20	
88.2% of original grouped cases correctly classified.	
79.1% of cross-validated grouped cases correctly classified.	

At the subspecies level (Table 5), 88.2% of the specimens were classified according to their predicted groups (79.1% on cross-validation), ranging at the group level from 70% to 100%. In this instance, there was some reciprocal misclassification in the original data between the three Gorilla subspecies, as well as between P. t. troglodytes subspecies, particularly between the central and eastern chimpanzees (P. t. troglodytes and P. t. schweinfurthii). One P. paniscus specimen classified with the P. t. troglodytes group. All H. sapiens molars classified correctly to the predicted group. On cross-validation, there was additional misclassification between G. b. beringei and G. b. graueri, but G. g. gorilla molars classified as before. Within Pan, P. t. verus and P. paniscus grouped as before, but there was further misclassification between P. t. troglodytes and P. t. schweinfurthii. Homo sapiens saw two molars classifying with the molars of P. t. verus (generally broader across the crown than the other subspecies) and one with P. t. troglodytes.

At the subspecies level, Function 1 accounted for 89.6% of variance and Function 2, 8%. Tooth size is the main discriminating feature, particularly relative breadth, with the relationship between the measurement across the buccolingual groove (negatively weighted along Function 1) and the measurement across the distal cusps (positively weighted along Function 1) playing a key role in discriminating between groups. Tables 4 and 5 present the classification accuracy of the 110 lower second molars at the species and subspecies levels.

Comparison of methods #1: DFA classification accuracy of mathematically-derived measurements versus anatomically-derived measurements, based on Pilbrow (2006)

The classification results at the species level, based on 110 specimens using variables derived from anatomically-based landmarks, are shown in Table 6. Classification accuracy ranges from 85% for G. gorilla and G. beringei, to 95% for P. paniscus, and 100% for P. troglodytes and H. sapiens. The overall classification accuracy is 94%, with cross-validation accuracy being 91%. Misclassified specimens of G. gorilla and G. beringei are reciprocally classified, and a single misclassified specimen of P. paniscus falls in H. sapiens.

At the subspecies level classification accuracy ranges from 50% for G. g. beringei, and 60% for P. t. schweinfurthii to 100% for P. t. verus and H. sapiens. The overall classification accuracy for subspecies is 84%, with a cross-validation accuracy of 76% (Table 7). Misclassified specimens are assigned to subspecies within the species, except for a single misclassified specimen of P. paniscus, which is assigned to H. sapiens.

Table 6 Classification accuracy of 110 LM2s at species level using anatomically-derived measurements.

	Species	G. beringei	G. gorilla	P. troglodytes	P. paniscus	H. sapiens	N	
Original	G. beringei	17	3	0	0	0	20	
G. gorilla	3	17	0	0	0	20	
P. troglodytes	0	0	30	0	0	30	
P. paniscus	0	0	0	19	1	20	
H. sapiens	0	0	0	1	19	20	
Cross-validated	G. beringei	15	5	0	0	0	20	
G. gorilla	3	17	0	0	0	20	
P. troglodytes	0	0	0	30	0	30	
P. paniscus	0	0	0	19	1	20	
H. sapiens	0	0	0	1	19	20	
93.6% of original grouped cases correctly classified.	
90.9% of cross-validated grouped cases correctly classified.	

Table 7 Classification accuracy of 110 LM2s at subspecies level using anatomically-derived measurements.

	Subspecies	G. b. beringei	G. b. graueri	G. g. gorilla	P. t. verus	P. t. troglodytes	P. t. schweinfurthii	P. paniscus	H. sapiens	N	
Original	G. b. beringei	5	3	2	0	0	0	0	0	10	
G. b. graueri	2	7	1	0	0	0	0	0	10	
G. g. gorilla	2	1	17	0	0	0	0	0	20	
P. t. verus	0	0	0	10	0	0	0	0	10	
P. t. troglodytes	0	0	0	1	8	1	0	0	10	
P. t. schweinfurthii	0	0	0	1	3	6	0	0	10	
P. paniscus	0	0	0	0	0	0	19	1	20	
H. sapiens	0	0	0	0	0	0	0	20	20	
Cross-validated	G. b. beringei	4	4	2	0	0	0	0	0	10	
G. b. graueri	3	6	1	0	0	0	0	0	10	
G. g. gorilla	3	1	16	0	0	0	0	0	20	
P. t. verus	0	0	0	9	0	1	0	0	10	
P. t. troglodytes	0	0	0	2	6	2	0	0	10	
P. t. schweinfurthii	0	0	0	1	4	5	0	0	10	
P. paniscus	0	0	0	0	0	0	19	1	20	
H. sapiens	0	0	0	0	0	0	1	19	20	
83.6% of original grouped cases correctly classified.	
76.4% of cross-validated grouped cases correctly classified.	

Tooth size is the main discriminating feature at the species and subspecies level. In both analyses, discriminant function one accounts for 89% of the overall variance and is heavily loaded by length and breadth dimensions and distance between cusps.

Comparison of methods #2: Mathematically-derived versus anatomically-derived DFA output based on Skinner et al. (2009)

The average classification accuracy of a DFA for a randomly-drawn sample of 25 specimens of P. t. troglodytes, P. t. verus and P. t. schweinfurthii using mathematically-derived measurements was 100% (original and cross-validated classification accuracy). This matches the 100% classification accuracy for 25 lower second molars reported by Skinner et al. (2009), based on a study of the EDJ of these teeth. The spatial groupings of the three species/subspecies output from the mathematically-derived DFAs (Fig. 7) closely match the groupings shown by Skinner et al. (2009).

Discussion

In the context of fossil hominin taxonomic studies, molars play an important role as these are the most abundantly preserved element in fossil assemblages and are well represented in museum and academic collections. However, many of these teeth are heavily worn or damaged on the crown, and researchers are usually forced to reduce limited samples of fossil specimens even further by discarding specimens without easily-identifiable anatomical features on the crown enamel. Some researchers have suggested approximating cusp peaks from the shapes of wear facets (Martinón-Torres et al., 2006), but this results in an analysis with a mixture of approaches to cusp-peak location.

Benazzi et al. (2011); Benazzi et al. (2012) have successfully avoided having to locate cusp tips on worn teeth, and have shown good taxonomic discrimination between worn Neanderthal and modern human teeth, using cervical crown outlines and occlusal crown outlines, which are still present in worn teeth. The present study differs from these studies, firstly in that 3D images are not required for the determination of cervical crown outlines: the surface enamel crown outline visible from 2D images is sufficient for the calculation of all landmarks used in this study, provided that the points where the grooves between cusps meet the outline shape are visible (or readily inferable) in the images used. This would include worn teeth up to and including wear stage 7 as defined by Smith (1984), wherein dentin is exposed on the entire molar crown surface, but the enamel rim remains largely intact; but not late wear stage 7 and wear stage 8, where the enamel rim is severely broken down. A second difference is that by using midlines between cusp grooves and midpoints to stand proxy for the mathematical centers of each cusp, rather than semi-sliding landmarks around the outline (Bookstein, 1996/7), it becomes possible to calculate the relative orientations of cusps through these cusp centers, both in relation to each other and to the longitudinal axis of the tooth. This study capitalizes on the fact that linear measurements and cusp angles, which have proven to be diagnostic measurements for taxonomic studies (e.g., Pilbrow, 2006, Pilbrow, 2007 and Pilbrow, 2010), are readily calculated from the landmarks located both around the perimeter and on the surface itself. Over and above analyses based on Euclidean coordinates of landmarks (GPA, PCA and EDMA –Euclidean Distance Matrix Analysis), this methodology allows for other types of analyses to be carried out that make use of raw measurements, including DFA, CV (coefficients of variation) analyses and other types of odontometric studies. The results of analyses based on these mathematically-derived measurements would then be useful for comparison with existing studies.

Figure 7 Convex hull plots of sample of 25 specimens chosen to match the species-subspecies groups used by Skinner et al. (2009).

The aim of the study was to establish whether mathematically-sited landmarks and the raw measurements derived from these, based on the occlusal crown outline of lower second molars, which is intact in both unworn and worn teeth, would lead to good separation in morphospace and high classification accuracy levels in a DFA. Further to this, the results should be at least equivalent in accuracy to the results achieved by other researchers. DFA classification accuracy outputs were compared to those achieved in two existing studies (modified from Pilbrow, 2006; Skinner et al., 2009).

PCA separation in morphospace and DFA classification accuracy

The results of the geometric morphometric and discriminant function analyses on 110 lower second molars of five species of extant hominoid (n ≥ 10 per subspecies) show good success in group separation, with 96.4% classification accuracy at the species level and 88.2% classification accuracy at the subspecies level. Specimens visualized on shape-only (shapespace) and size-and-shape (formspace) PCA plots grouped according to morphological differences that are diagnostic for each species and genus. Shape differences were observed between Gorilla, Pan and Homo sapiens at the genus level in the shape-only analysis, but as expected in a shapespace plot, there was no sexual dimorphism evident between molars belonging to male and female gorillas, and no interspecific general shape differences between P. paniscus and P. troglodytes (Singleton et al., 2011), with the exception of specimens of P. t. verus, which are on average relatively wider across the crown than other subspecies (Uchida, 1996; Pilbrow, 2006; Dykes, 2018). There was some overlap between P. t. troglodytes and P. t. schweinfurthii molars in shape space, which is to be expected as these two subspecies interbred until relatively recently (Hey, 2010; Gonder et al., 2011). Shape variation in H. sapiens lower second molars exceeded that of other species, undoubtedly a factor of differential evolution in molar cusp simplification (reduction in size, or absence of a hypoconulid), due to regional differences in basic subsistence strategies of hunter-gatherers and agriculturalists over many millennia (Bailit & Friedlaender, 1966; Brace & Mahler, 1971; Sofaer, 1973; Brace, Rosenberg & Hunt, 1987; Corruccini, Potter & Dahlberg, 1983; Corruccini, 1984; Larsen, 1995; Dempsey & Townsend, 2001; Grine, 2002; Grine, 2005; Brown & Maeda, 2004; Pinhasi, Eshed & Shaw, 2008; Emes, Aybar & Yalcin, 2011; Hodder, 2017; Ungar, 2017; Dykes, 2018). This unique shape variation in modern H. sapiens will be examined in detail in a further study, based on larger sample sizes.

When size is added to the analysis, there is excellent spatial separation at the species level on a PCA formspace plot. Gorilla molars were reasonably separated between the two species represented, with sexual dimorphism being observed between molars belonging to males and females at the species and subspecies level. Eastern lowland gorillas, G. b. graueri, have the largest body size of all gorillas (Jungers & Susman, 1984), and this is reflected in the grouping of their molars, towards the positive extreme of the x-axis in the plot. Western lowland gorillas, G. g. gorilla, grouped at lower values along the x-axis, with only two male specimens overlapping with males of the Eastern mountain gorillas, G. b. beringei. Gorilla gorilla as a species showed more shape variability along PC2 in their lower second molars than G. beringei, with some molars that are relatively narrow buccolingually belonging both to males and females grouping at higher values along the y-axis. On the other extreme of the x-axis, P. paniscus grouped cohesively in the top left-hand quadrant, having the smallest molars that are generally relatively narrow. Most P. troglodytes molars grouped above the x-axis, in a similar range along PC2 as that observed for Gorilla species with relatively narrow molars; however, the broader molars of P. t. verus fell below the x-axis, in the quadrant occupied by H. sapiens molars. Overall, groupings in morphospace followed expected patterns of molar shape and size differences between genera, species, subspecies, sexes and subsistence strategies.

Comparing the PCA and DFA results, it can be seen that in both the shape-only (shapespace) and the shape-and-size (formspace) PCA analyses, there are similarities between the factors affecting the first two principal components of the PCAs on the one hand, and the factors accounting for the main canonical loadings along the first two functions of the DFAs on the other. The relative warps of the wireframes traced from the negative to the positive end of the axes of the PCA plots show a shape change between relative breadth of tooth (both axes), the extent of the buccal development groove (x-axis), the amount of curvature of the hypoconulid (y-axis), and along both axes, a change in the relative dimensions (raw measurements and orientations) between the distal cusps and other breadth and length variables. While size is shown to be an important discriminating factor between groups, relative dimensions and angles between variables therefore also play significant diagnostic roles. The methodology of landmarking worn teeth discussed in this paper makes provision for landmarks to be placed not only at selected inflexion points around the outline of the tooth but also on the enamel surface itself, irrespective of how worn that might be, thus providing for the inclusion of key cusp-related data (dimensions and angles) into the analysis that would otherwise be lost, if only perimeter-shape landmarks are to be used.

Comparisons of accuracy between mathematically-derived measurements and anatomically-derived measurements

Mathematically-derived measurements compare well with anatomically-derived measurements in both comparative analyses, based on classification accuracy outputs from stepwise DFAs. In the first analysis of 110 specimens from five species and eight subspecies, using only nine variables, the mathematically-derived measurements produced a classification accuracy of 96.4% at the species level and 88.2% at the subspecies level, versus 93.6% at the species level and 83.6% at the subspecies level for the anatomically-derived measurements taken from the same 110 teeth. In the second analysis of 25 specimens from P. t. troglodytes (n = 5), P. t. verus (n = 10) and P. paniscus (n = 9), the 100% classification accuracy of lower second molars based on anatomically-based landmarks on the EDJ was matched with 100% classification accuracy of lower second molars based on mathematically-derived measurements on and around the OES. The improvement in classification accuracy compared to the previous comparison of 110 specimens is an artefact of sampling: smaller sample sizes and a smaller subset of taxa were used to match those used in Skinner et al. (2009). In particular, the exclusion of P. t. schweinfurthii, which overlaps in molar size and shape with P. t. troglodytes improves classification accuracy. Given this identical result in classification, the mathematical landmarking method might therefore provide a means to assess how well the EDJ taxonomic signal relates to that of the OES, as discussed by Skinner et al. (2009), providing a solution to both of the frustrations experienced by this group of researchers—the scarcity of unworn teeth in museum collections and the problem of the need for increased subjectivity in the placement of landmarks on a worn OES, as compared to the well-defined ridges of the EDJ. Since landmarking only takes a few minutes to complete, this might enable quick verifications to be made, confirming the results of the EDJ analyses.

The results from the comparative analyses provide a reasonable level of comfort that the mathematically-based methodology of landmarking the crown surface might readily replace more traditional methods based on anatomical landmarks, to enable worn teeth to be included on an equal basis with unworn teeth in morphometric analyses. Worn teeth might include individual specimens of importance (particularly fossil hominin holotypes or paratypes) and important groups (e.g., modern human hunter-gatherer groups, whose teeth tend to be more heavily worn than groups with soft diets). Inclusion of worn teeth allows for the augmentation of poorly-represented fossil hominin species and for the balance between groups to be improved (e.g., between males and females, or between different extant populations). Provided that the outline shape and the five cusp intersections are identifiable on the image, landmarking is possible, irrespective of the lack of sharpness of features on the surface. Although the present study necessarily made use of relatively unworn teeth, to enable comparison of classification accuracy with previous studies that used unworn teeth, further papers are being prepared where the methodology is applied in classifying worn fossil hominin and modern human molars. The methodology is designed to be adaptable for all extant hominoid and extinct hominin molars, for all types of analysis involving landmarks or measurement variables, and for all levels of group comparison at the level of genus, species, subspecies, population and sex.

Conclusions

Enamel crown surfaces of lower second molars provide reliable data for use in extant hominoid systematics studies, and by extension, they are reliably applied to extinct hominin taxonomic studies as well. Diagnostic features of the enamel surface of molar crowns include the general proportions of molars, size, occlusal outline shapes and the dimensions and orientations of cusps, both in relation to each other and to the longitudinal axis of the molar. In this study, the mathematical placement of landmarks not only around the perimeter outline of the molar but also at cusp centers and midlines allows for shape and size analyses, as well as analyses that make use of cusp angles and dimensions. Euclidean coordinates may be used directly in geometric morphometric studies, or distances and angles between landmarks may be calculated for analyses using raw measurements.

This study seeks to establish whether strategically-sited mathematically-derived landmarks on and around the enamel surface of molars produce good discrimination between groups in morphospace (at the species and subspecies levels) and classification accuracy that is at least as high as that derived from using traditional, anatomically-derived measurements and landmarks. The method was tested on 110 lower second molars representing five hominoid species (eight subspecies). Good separation of groups in morphospace was achieved after a generalized Procrustes analysis in a shapespace (shape-only) principal components analysis. This grouping was further improved by adding size as a variable into the analysis, in a formspace (shape-size) principal components analysis, this time allowing for visualization of sexual dimorphism between male and female gorillas and for separate groupings of bonobo and common chimpanzee molars. Classification accuracy of 96.4% at the species level and 88.2% at the subspecies level was achieved in a discriminant function analysis, using only nine linear and angular measurements. Comparing the classification accuracy of these mathematically-derived measurements to the traditional, anatomically-based methodologies used in two existing studies (Pilbrow, 2006; Skinner et al., 2009), the results are identical (as compared to Skinner et al., 2009) or marginally improved (as compared to the type of methodology described in Pilbrow, 2006), providing comfort that the methodology may confidently be used as an alternative to methodologies relying entirely on identifiable anatomical features. Worn molars may be therefore be included equally confidently as unworn or moderately-worn molars in geometric morphometric and other statistical studies, even if the enamel surface is severely damaged, but the perimeter edge of the tooth and the cusp intersections are still clearly identifiable. This is crucial for studies where sample sizes are low, such as in analyses involving fossil hominin molars; indeed, in some cases, the sample, usually already limited, includes holotypes or proxies for holotypes with extremely worn molars. The same advantage holds when an imbalance exists in the availability of relatively unworn teeth between males and females in a sample, between geographical groups, subspecies, or diet-based groupings (where some groups may have heavily worn teeth but others might not).

The methodology described in this study is rapid, requires little or no subjectivity or expertise, and can be carried out using readily available software. We conclude that this methodology provides results that are equally as accurate as methodologies based on anatomical landmark sites, and that it can be recommended for augmentation of sample sizes in studies involving worn teeth.

Supplemental Information

File S1 Specimen list and raw Euclidean landmark coordinates, 110 specimens

Click here for additional data file.

File S2 Summary measurement statistics per species and subspecies using mathematically-derived measurements in DFA

Click here for additional data file.

File S3 Summary output - canonical functions - DFA output based on mathematically-derived measurements

Click here for additional data file.

File S4 Summary output of DFA based on anatomically-derived measurements

Click here for additional data file.

File S5 DFA canonical outputs based on anatomically-derived measurements

Click here for additional data file.

The authors are grateful for the comments and suggestions provided by reviewers of this paper. Susan J. Dykes extends sincere thanks to the following museums/collections for access to modern human dental material: The Raymond Dart collection at the University of the Witwatersrand, Johannesburg; the Iziko museum, Cape Town; the Musée de l’Homme, Paris; the Duckworth collection at Cambridge University. Varsha C. Pilbrow acknowledges the following museums for access to chimpanzee and gorilla dental material: American Museum of Natural History, NY; Anthropologisches Institüt und Museum der Universität Zürich-Irchel, Zürich; British Museum of Natural History, London; Field Museum of Natural History, Chicago; Museum of Comparative Zoology, Harvard; Muséum National d’Histoire Naturelle, Paris; Powell-Cotton Museum, Kent; Peabody Museum of Anthropology, Harvard; United States National Museum, Washington, D.C.; Musée Royal de l’Afrique Centrale, Tervuren; Zoologisches Museum, Berlin; Anthropologische und Zoologische Staassammlung, Münich.

Additional Information and Declarations

Competing Interests

Author Contributions

Data Availability

The authors declare there are no competing interests.

Susan J. Dykes conceived and designed the experiments, performed the experiments, analyzed the data, prepared figures and/or tables, authored or reviewed drafts of the paper, approved the final draft, database of images.

Varsha C. Pilbrow conceived and designed the experiments, performed the experiments, analyzed the data, authored or reviewed drafts of the paper, approved the final draft, database of images.

The following information was supplied regarding data availability:

Raw data is available in File S1. This includes all the landmark coordinates of the 110 specimens analyzed in this study.The SPSS output (summary statistics and correlation coefficients data) is available in Files S2 and S3 for the mathematically-based method and Files S4 and S5 for the comparative anatomically-based method.

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
