# Peer review of "A mathematical landmark-based method for measuring worn molars in hominoid systematics"

_PeerJ, doi:10.7717/peerj.6990_

## Round 0.1 · original submission · Minor Revisions

I was very pleased to edit this manuscript having had personal research experience in losing data due to worn teeth. The reviewers and I concur that the contribution to the field is robust and the manuscript is very well-prepared. There are a few lingering issues that will be very simple to address in a revision. At this point, given the quality of the submission and the highly positive reviewer comments, I will not need to send a revision out to review for a second time but would like to see the concerns and requested clarifications addressed before accepting for publication. Please be careful to address specific reviewer comments in your revision and rebuttal. Thanks for submitting to PeerJ and congratulations!

·

Basic reporting

This manuscript is well-written and very clear. There is sufficient background and relevant literature is cited. The authors have succeeded in creating a clear and concise argument.

The raw data are provided; however, it was difficult for me to tell which data was from the mathematically derived method and which data were from the anatomically derived method.

Experimental design

The design was clear and well described. Given the raw data and description of statistical methods, replication could easily be achieved. The figures of the landmarks were also well constructed and clear. Dental wear is an issue with all dental anthropological studies, this research addresses this problem and provides possible concrete solutions to assessing morphology even with advanced occlusal wear. Their focus was on molars, but an extension to other teeth could be imagined.

Validity of the findings

While the mathematical model did not perform as well in the DFA as the anatomical method, its validity is still presented. The sample size is small, but general trends can still be identified. In discussions of the DFA, the authors may want to address sample size and the number of variables input into the model. There could be an issue of overfitting with the sample. A step-wise analysis was performed, but the number of variables in the final model was not indicated. Also, results could be discussed in terms of positive predictive value as opposed to 'correct classification.'
Finally, some discussion of sexual dimorphism could be included. While teeth were scaled and size was not generally evaluated, there could be sexual dimorphism present within the sample. Some discussion of dental sexual dimorphism or a simple statistical test of dimorphism could be included.

Additional comments

Thank you for the opportunity to review this manuscript. It is very well-written and clear. The methods are well described and could readily be used in future research projects on worn teeth. There are only minor comments for improvement on what is already an excellent manuscript.

·

Basic reporting

The paper is very well written with only a few grammatical errors (see below for the few that I noticed). Figure 1 is especially useful in helping to understand the landmarks that the authors collected data on. Overall, I would strongly support this paper being published.

Grammatical edits:
1. On line 264, add 'and' before 'orientation'.
2. On line 383, add 'to' before '95%'.
3. On line 384, change 'cross-validated to 91%' to 'with cross-validated accuracy being 91%'.
4. On line 387, add 'and' before '60%'.
5. On line 425, change 'This' to 'The'.
6. On line 441, change 'the' to 'this'.
7. On line 530, change 'to' to 'with'.

Experimental design

Based on the results provided (especially the high classification accuracy achieved both at the species and subspecies level) the innovative and creative methods developed by the authors have great potential to be used to document the morphology of worn molars in taxonomic and phylogenetic studies of extant and fossil hominoids and, thus, could significantly increase the number of specimens available for analysis, as the authors appropriately note in their discussion and conclusion. Moreover, the methods require little in the way of technology and are easily replicated (with little subjective decision-making required in the placement of landmarks), making them more readily available to researchers with fewer resources and/or institutional support. Consequently, this submission has the potential to be extremely impactful in the field of dental anthropology and hominoid systematics given how analyses of molar morphology generally do not include well worn molars, which can substantially lower sample sizes. However, there are a number of issues that, if addressed, would improve the manuscript further (see below).

1. On line 90, I would rephrase the sentence that mentions Type I landmarks to read 'These anatomical landmarks, which are classified as Type I (Dryden & Mardia, 1998)...' Also, I believe that Bookstein, 1990 was the author that initially developed the system of classifying landmarks into types I-III.

2. While I understand the authors' reasoning for using relatively unworn molars in this study, it would have made their study even stronger to provide some proof of concept that worn molars can also be accurately classified using these methods. I understand that the landmarks they collected data on are unlikely to be affected by wear, but it is a bit odd to be arguing for the use of these methods for classifying worn molars when no worn molars were included in the analysis. I realize that this may entail a substantial amount of work and if the authors (and editor) think that this is unnecessary for this paper, I would suggest adding a line or two about conducting a follow up study in the discussion and/or conclusion.

3. I am not clear how landmarks 21-25 necessarily mark the 'mathematical center of each cusp' given that landmark 1 is in the mathematical center of the molar and, thus, will not be on the margin of each cusp (i.e., the line drawn from landmark 1 to landmarks 16-20 may include parts of other cusps or only a portion of the cusp in question depending on where landmark 1 is placed and, consequently, landmarks 21-25 will not be at the center of the cusp in most cases). I would argue for naming those landmarks using a different phrase.

4. Can the authors provide a brief explanation of why they selected the nine measurements they decided to include in this analysis and not other measurements?

5. From the discussion of the methods, it is not entirely clear whether the data collected for the anatomically-based methods were collected by the authors on the same specimens that they collected the data using mathematically-based methods on. I believe they were, but this should be made clearer in the manuscript.

6. I would argue for changing the names of measurements 2 and 3 in Table 3 to 'Buccolingual breadth at the mesial and distal cusps'. Also, should the names of measurements 5 and 6 start with 'Length of'? In the description of measurements 7 and 8, it appears that the authors are stating, for example (for measurement 7), landmarks 21 through 25 when I believe they mean to say landmarks 21 and 25 (there is a similar issue throughout the descriptions of these landmarks). Finally, the description of measurement nine is not clear to me.

7. I would suggest adding a legend to figures 5 and 6 denoting in one box what taxon each of the symbols represents. Also, the authors should orient the reader as to which direction on the wireframes is mesial, distal, etc.

(see next section for additional suggestions/comments)

Validity of the findings

8. Have the authors examined their data to determine if any four or six cusped individuals are outliers? One would expect that if this method accurately documents morphological differences within taxa for molars with different numbers of cusps that they would be clearly separated from others in their group. The authors even state on lines 468-474 that one of the reasons that shape variation was greater in humans than in great apes in this study was the absence of a hypoconulid in some human specimens, but did they assess whether these teeth were significantly different in shape using their landmark data?

9. With respect to the inter- and intra-observer error studies, can the authors provide more information? First, they cite the maximum differences in tilt angles along each plane, but do not discuss the extent to which this might have affected the placement of landmarks and, thus, their results. Was there a study done to examine this latter issue? Second, while they provide data on the mean deviation in the x and y axis for landmarks, they do not discuss the extent of intra-observer vs. inter-observer error. Also, these data should be contextualized. For example, the authors could note how large the error was compared to the distances between landmarks or compared to the distance differences between individuals in a single taxon. In other words, could this error have impacted the results of this study (granted, this seems unlikely given how small the error appears to be)?

10. I am confused as to how the classification accuracy of the mathematically-based methods for the comparison with Skinner et al.'s data was so much higher than it was when the larger sample was compared to Pilbrow's data (especially given that data were collected on the same variables and same taxa). Can the authors discuss this?

Additional comments

Minor points:
1. I would argue for reporting the classification accuracy with cross-validation in the abstract as that is the more statistically valid measure.
2. While I realize that it may not be technically inaccurate, I would prefer the use of 'taxonomic' rather than 'taxonomical'.
3. I would change 'populations' in line 192 to 'individuals'.
4. Shouldn't 'subjectivity' on line 524 be 'objectivity' instead?
5. In table 2, a space should be added between the information for landmarks 27 and 28.

---

## Round 0.2 · accepted · Accept

I am delighted to the accept the revised manuscript for publication!

#